# Rapamycin-Reactivated Lipid Catabolism in *Eruca sativa* Mill. Exposed to Salt Stress

**DOI:** 10.3390/cells14141083

**Published:** 2025-07-15

**Authors:** Emilio Corti, Sara Falsini, Gianmarco Patrussi, Nadia Bazihizina, Cristina Gonnelli, Alessio Papini

**Affiliations:** Department of Biology, University of Florence, Via Pier Antonio Micheli 1–3, 50121 Firenze, Italy; emilio.corti@unifi.it (E.C.); sara.falsini@unifi.it (S.F.); gianmarco.patrussi@edu.fi.it (G.P.); nadia.bazihizina@unifi.it (N.B.); cristina.gonnelli@unifi.it (C.G.)

**Keywords:** salt stress, *Eruca sativa*, autophagy, rapamycin, peroxisomes, glyoxysomes, pexophagy

## Abstract

Salt stress is one of the most common factors reducing the productivity of crops. We tested the effect of Rapamycin, an mTOR inhibitor and autophagy inducer, for the possible amelioration of high-salinity stress in *Eruca sativa*. We analyzed the germination rate, the macro- and micro-morphology of seedlings, and the ultrastructure of cotyledons with a Transmission Electron Microscope. The most striking observation was that salt stress blocked the catabolism of the lipid droplets stored in the embryos of *E. sativa*, also dramatically reducing the starch storage capability in the plastids. As a consequence, lipid droplets remained in the developing seedlings until a late stage. On the contrary, the catabolism of the lipid storage in the embryos in the presence of rapamycin and salt stress was comparable to the control, even if the starch stored in the plastids was lower. Rapamycin-induced autophagic activity was shown by characteristic ultrastructural changes, such as increased membrane recycling. Part of this activity was interpreted as pexophagy, i.e., the autophagy of peroxisomes, where an increase in their turnover rate could be necessary to maintain an active glyoxylate cycle.

## 1. Introduction

Crop production is negatively affected by abiotic stresses, such as salinity, which is one of the main factors that compromise plant growth, development, and productivity [1,2]. The excessive occurrence of salt in soil results in long- and short-term plant impairment. In the short term, salinity leads to osmotic stress and the alkalization of vacuoles, resulting in an overall reduction in water uptake, affecting plant–water relations and cell expansion [3]. In the long term, salinity results in ion toxicity because of the high accumulation of salts in plant tissues [4]. Comprehensively, salt stress leads to several disorders concerning photosynthesis, oxidative stress, water status, and enzyme activity, which induce alterations in plant germination, growth, and development [5,6,7].

In the life cycle of cultivated plants, germination and seedling development play a crucial role that influences their final productivity and, thereby, their crop yield [8]. In salt-treated plants, the limited water absorption and ion accumulation affect many biochemical processes, impairing lipid and protein reserve mobilization [9,10,11]. Moreover, salt stress influences plant cell structure at every developmental stage, particularly in seedlings [12]. Modifications of leaf anatomy, such as variation in the mesophyll and epidermis thickness and changes in the intercellular space size, were recorded in salt-affected plants, showing contrasting responses based on the species and salt-tolerance ability [13,14]. Lastly, salinity induces changes in cell organelles, such as chloroplasts, mitochondria, and plasma membranes, as well as triggers autophagic processes [15]. Salinity (sodium ions) would not necessarily affect organelles directly, but it would induce K+ loss from chloroplasts in both glycophytes and halophytes [16], leading to a change in the organellar dimension. Particularly relevant is the role of peroxisomes since the oxidative nature of this organelle’s metabolism makes it sensitive to stress [17]. Peroxisomes would increase in number in the presence of salt stress [18]. Pexophagy, the selective macroautophagy of peroxisomes, is the process by which the plant cell recycles obsolete peroxisomes [16,19]. As reported by Pu et al. [20], the occurrence of autophagy fosters plant resistance against a series of stresses, including salt stress. The autophagic process allows the transport of harmful or damaged cellular components into lytic vacuoles to be recycled, thus maintaining cellular homeostasis [21]. To regulate autophagy, eukaryotic organisms feature a serine/threonine protein kinase called the Target of Rapamycin (TOR) [22]. Under abiotic stresses, e.g., a nutrient deficiency, salt stress, and drought, the inhibition of TOR activity triggers the activation of autophagy, mitigating negative stress-induced responses [23]. Among the known TOR inhibitors, the fungal antibiotic Rapamycin is commonly used to block the activity of such a complex and to study the enhanced autophagy [24,25].

Given the above considerations, in this study, we investigated the effect of Rapamycin on *Eruca sativa* L. seedlings. By assessing salt-induced changes in leaf anatomy and cell ultrastructure, our aim was to determine whether Rapamycin could mitigate the effects of salinity by enhancing autophagic processes in order to identify novel strategies to counteract salt stress during seed germination and early seedling development.

## 2. Materials and Methods

### 2.1. Seed Germination Set-Up

A ready-made solution of Rapamycin, 2.5 mg/mL (2.74 mm in dimethyl sulfoxide (DMSO), from *Streptomyces hygroscopicus*, Sigma-Aldrich^®^, now Merck, Saint Louis, MO, USA), hereafter referred to as Rapamycin, was used to prepare the final culture medium.

Rapamycin was provided with DMSO as a solvent. This substance was used in the experiment because of its possible non-neutral effects: DMSO was evaluated as possibly phytotoxic in rice seedlings [26], while in barley and wheat [27], it showed other biological effects not related to toxicity (at the concentrations used in our experiment). The DMSO concentration was that derived from the amount supplied together with Rapamycin.

To evaluate the germination response to the Rapamycin supply under salt treatment, *E. sativa* Mill. seeds (Blumen^®^, Doha, Qatar) were treated with distilled water (as a control); NaCl 1.2% *w*/*v*; DMSO 0.2% *v*/*v* (as a further control); DMSO 0.2% *v*/*v* + NaCl 1.2% *w*/*v*; Rapamycin 5 µM (supplied in DMSO 0.2% *v*/*v*); and Rapamycin 5 µM (supplied in DMSO 0.2% *v*/*v*) + NaCl 1.2% *w*/*v*. The DMSO treatment was added at the same concentration since Rapamycin was provided in a solution with DMSO 0.2% *v*/*v*. The concentration of 5 µM for Rapamycin was chosen according to previous experiments in the literature, such as those in Deng et al. [28], who stated that 5 µM is the saturating concentration for Rapamycin. The choice of the concentration of NaCl, i.e., 1.2%, was made according to previous tests carried out by the authors on *E. sativa*. The treatments were carried out for 96 h in a thermostatic chamber under conditions of saturated humidity, with a temperature of 21 °C, a photoperiod of 18/6 h (light/darkness), and light radiation of 200 µmol m^−2^s^−1^.

Before treatment, the seeds of *E. sativa* were sterilized in 70% (*v*/*v*) ethanol for 20 min and washed three times with distilled water. Then, 10 seeds were placed in 9 cm diameter Petri dishes on filter paper moistened with 1 mL of treatment solutions for each seed and put inside the thermostatic chamber. A total of 1 mL of water was added after 48 and 96 h. Each treatment was conducted in triplicate.

### 2.2. Germination and Seedling Assessment

The germination rate was evaluated by counting the number of seeds with a rootlet at least 2 mm long. This rate was recorded every 12 h up to 72 h (3 days). At each time point, the number of germinated seeds with roots was counted. At 72 h, the seedlings from each treatment were collected to measure stem and root lengths using graph paper. Stem length was defined as the distance from the base of the cotyledons to the beginning of the root. Moreover, the effect of the treatment on root and stem lengths was calculated using the following formula: (1 − (length of treated sample/mean control length) × 100).

### 2.3. Light Microscopy and TEM Microscopy Observations

Samples from the cotyledons of each treatment were collected and fixed in 1.25% (*v*/*v*) glutaraldehyde in 0.1 M phosphate buffer (pH 7.2), stored at 4 °C for 24 h, and then post-fixed in 1% OsO_4_ in 0.1 M phosphate buffer (pH 7.2). After a graded ethanol and a propylene oxide step, the samples were embedded in Spurr epoxy resin [29].

Semithin and ultrathin cross-sections were obtained by a Reichert-Jung (Heidelberg, Germany) ULTRACUT ultra-microtome with a diamond knife. Semithin sections were stained with 0.1% toluidine blue and observed with a Leitz DM-RB—Fluo Optic microscope (Leitz-Wetzlar, Wetzlar, Germany) equipped with a digital camera (Nikon DS-L1, Tokyo, Japan).

Ultrathin sections, placed on copper grids, were stained with uranyl acetate [30] and subsequently with lead citrate [31]. Images were acquired using a Philips CM12 Transmission Electron Microscope (Philips, Amsterdam, The Netherlands).

Both light microscopy images and TEM images were analyzed with ImageJ2 (version 2.16.0) (https://imagej.net/ij/ (accessed on 1 July 2025)). Light microscope images were analyzed with ImageJ to calculate the area occupied by lipid droplets and to measure various macromorphological traits of the seedling. Fifteen measurements per plant were taken, with three replicates (three plants, for a total of 45 measurements) for each treatment.

### 2.4. Statistical Analysis

GerminaQuant for R (https://flavjack.shinyapps.io/germinaquant/ (accessed on 2 July 2025)) was used for the germination analysis of *E. sativa* seeds. The germination progression of the different treated groups was assessed, together with the calculation of the germination percentage, defined as:$$GRP=\left(\frac{\sum_{i=1}^{k}n1}{N}\right)\;*\;100$$
where n1 is the germinated seed number in the *i*th time, and k is the last day of the evaluation process for germination.

The statistical analysis of root and stem length, intercellular space, cotyledon thickness, lower epidermis thickness, upper epidermis thickness, palisade mesophyll area, spongy mesophyll area, and lipid droplet percentage was performed with GraphPad Prism (version 8.0.1) using a one-way ANOVA (one-way Analysis of Variance) and the Tukey–Kramer test for multiple comparisons.

## 3. Results

### 3.1. Effects on Seed Germination

Figure 1 shows the effect of the treatment on *E. sativa* germination. The seeds treated with water (external control), DMSO only (control for this experiment), and Rapamycin did not show any significant difference in the final percentage of germinated seeds. Although the DMSO- and Rapamycin-treated seeds began germination with some delay compared to the water-only group, after 24 h, the number of germinated seeds reached the same level (about 100%). A marked delay in the onset of germinated seeds was observed for the seeds treated with NaCl, DMSO + NaCl, and Rapamycin + NaCl compared to the other groups, as well as a significant reduction in their germination percentage. The seeds treated with NaCl and Rapamycin + NaCl had a significantly lower germination percentage than the water, DMSO, and Rapamycin groups, but a higher germination percentage than the seeds treated with DMSO + NaCl.

### 3.2. Root and Stem Length

The morphological aspect of the plantlets exposed to the different treatments was recorded at the end of the experiment (Figure 2A). The root and stem lengths were reduced in the seedlings treated with Rapamycin + NaCl and DMSO + NaCl compared to the control groups, i.e., water alone and a solution of DMSO only (Figure 2B,C). However, the Rapamycin + NaCl group showed a smaller reduction in stem length compared to the DMSO + NaCl group. Moreover, the plants in the Rapamycin group showed stem length values that were significantly higher than all the other groups (Figure 2C).

### 3.3. Fresh and Dry Weight

A reduction in dry weight due to salt treatment was detected in the DMSO + NaCl group, while no decrease in this parameter was recorded for Rapamycin + NaCl compared to Rapamycin alone. On the other hand, no significant effects of salt stress were observed on the fresh weight parameter (Figure 3A,B).

### 3.4. Cotyledon Anatomy with the Light Microscope

Semithin sections of the cotyledons (Figure 4) showed a significant reduction in plastid number and dimension in the seedlings treated with NaCl (Figure 4D) and DMSO + NaCl (Figure 4E) compared to those treated with water, DMSO only, and Rapamycin (Figure 4A–C, respectively). In the latter treatment, there was also a marked increase in lipid droplets. In the samples from the control (Figure 4A) and Rapamycin (Figure 4C) treatments, well-formed plastids without lipid accumulation were observed. The situation of the Rapamycin + NaCl treatment (Figure 4F) was similar to the control, although with lower starch storage in the plastids.

Regarding the intercellular spaces, the Rapamycin-treated seedlings showed the highest values, with significant differences compared to the DMSO-only and DMSO + NaCl treatments (*p* < 0.05) (Figure 5A). Regarding cotyledon thickness, the slides of the Rapamycin + NaCl group showed a higher thickness compared to the Rapamycin alone group (*p* < 0.05), while the DMSO + NaCl group showed a reduction in this parameter compared with the DMSO-only control group (*p* < 0.05) (Figure 5B).

The upper (Figure 6A) and lower (Figure 6B) epidermis showed a significant increase in thickness in the salt-treated plants of the Rapamycin group compared to the plants of the DMSO group.

Regarding the palisade mesophyll area (Figure 7A), a reduction was observed in the seedlings treated with DMSO + NaCl compared to the DMSO-only control group (*p* < 0.05), while no significant differences were recorded between the Rapamycin groups (Figure 7A). Hence, Rapamycin, in the presence of NaCl stress, restored a palisade mesophyll area comparable to that of the control. In addition, the spongy mesophyll area also showed an increase in the Rapamycin + NaCl treatment compared to the Rapamycin alone treatment and when compared to the DMSO treatments (Figure 7B).

Seedlings grown in DMSO + NaCl showed a section area occupied by lipid aggregates within the cell about 400% higher than all other treatments (*p* < 0.05). A low occurrence of lipid aggregates was instead observed in seedlings treated with Rapamycin alone and Rapamycin + NaCl (Figure 8).

### 3.5. Ultrastructure of the DMSO + NaCl- and Rapamycin + NaCl-Treated Mesophyll Cells

The observations showed a large amount of lipid droplets and electron-dense bodies apparently similar to phenol compounds in the vacuole with the DMSO + NaCl treatment (Figure 9A,B,D,F).

Most of the chloroplasts showed no well-defined grana stacks or grana formed by only a few series of membranes. Thylakoid swelling occurred, and several chloroplasts exhibited a less electron-dense area within the stroma region (Figure 9C,E). Moreover, the occurrence of starch grains inside the chloroplasts was rare.

The samples from the Rapamycin + NaCl treatment showed numerous plastids containing starch grains. Few lipid droplets and no phenolic compounds were present (Figure 10A–D,F). On the contrary, some vesicles and multivesicular bodies (MVBs) could be recognized inside the vacuoles (Figure 10E). Most of the chloroplasts showed defined grana and no thylakoid swelling.

## 4. Discussion

The negative effect of DMSO in combination with NaCl on the seed germination in *E. sativa* seeds observed here, from the point of view of lipid use by the growing seedling, was somewhat unexpected. This effect was more pronounced than that of NaCl alone. DMSO is a solvent used to solubilize several otherwise poorly soluble polar and nonpolar molecules [32], such as Rapamycin itself. However, the genotoxic effects of DMSO on the root tip cells of *Vicia faba* were reported by Valencia-Quintana et al. [33], although at much higher concentrations (20–40%) than those employed here. Apparently, DMSO is able to increase membrane permeability [22], possibly enhancing NaCl uptake by plant cells. In the same DMSO + NaCl combination, but also with the NaCl treatment alone, the lipid droplets in the cytoplasm remained stable during germination. Normal germination occurs through the glyoxylate cycle, which plays a crucial role by allowing the synthesis of sugars, using acetyl-CoA produced by the beta-oxidation of non-esterified fatty acids [34]. The failure of lipid catabolism in the presence of salt stress has also been observed in the unicellular algae *Botryococcus* [35] and *Chlorella* [36], as well as in Arabidopsis [37] and *E. sativa* [10]. In *E. sativa* cotyledons, the lipid droplets derive from those already present in the seed embryo; therefore, their accumulation (or lack of degradation) should be attributed to the reduced efficiency of the lipid catabolism pathway. Lipids represent the main carbon source here, and an impairment of lipid metabolism leads to carbon starvation, which persists until fully formed leaves develop. Salt stress is known to modify the expression of proteins related to lipid metabolism in *Mesembryanthemum* [38], although the affected proteins are mainly involved in fatty acid synthesis. In Arabidopsis, isocitrate lyase, a key enzyme of the glyoxylate cycle, has been linked to salt stress, supposedly acting as a protective factor [29]. In *E. sativa*, the observed increase in glyoxysomes (as defined in [39]) during salt stress [10] might be due to increased enzyme production, since isocitrate lyase (and other enzymes of the glyoxylate cycle) is localized in the peroxisomes [40].

From the point of view of the expression of transcription factors, Franzoni et al. [41] observed an increase in the expression levels of DtRD29A and DtHB7, known markers of salt and drought stress, the latter being a transcription factor of the HD ZIP family and a negative regulator of ABA signaling by acting as a positive regulator of protein phosphatase (PP2C) genes. Huang et al. [42] observed an upregulation of the expressed genes involved in arginine and proline metabolism and alpha-linolenic acid metabolism and a downregulation of carbon fixation and protein synthesis in seedlings of Eruca after germination. The decrease in starch production observed here with a high concentration of NaCl could be explained by these changes in gene expression. However, we must keep in mind that salt stress affects multiple points of metabolism, and hence, some results (such as the increase in lipid droplets) may derive not from the direct toxicity of salt, but from several processes directly or indirectly affected by the high concentration of NaCl, including water movement between the inside and outside of the cytoplasm or organelles.

One hypothesis about how Rapamycin influences phenotypes is described as follows. Rapamycin is a multifaceted drug with a specific action on the mTOR signaling pathway and is also able to cause cell cycle arrest, but it has also been used as an anti-fungal agent and as an immunosuppressant [43], implying that its mechanisms of action are probably multiple and not well understood in plants. Our results showed that the administration of Rapamycin was able to reestablish a normal amount of lipid droplets, even in the presence of salt, suggesting that autophagy may play a role in maintaining lipid catabolism during seed germination under salt stress. This result is novel in the literature. Rapamycin-induced autophagic activity might increase the turnover of peroxisomes, thereby restoring the capability to utilize lipid resources. The recovery would be partial since little starch accumulates in plastids, even in Rapamycin-treated plants. An interpretation of the results in graphical form is shown in Figure 11 from the point of view of peroxisome recycling, although with changes in cell size. Rapamycin-induced autophagic activity of peroxisomes has been defined as pexophagy [44], whose function is the elimination of malfunctioning peroxisomes during seed germination [45]. Rapamycin is also able to slow down cell proliferation [46], which might be one of the reasons for the increase in intercellular spaces observed in the Rapamycin-treated cotyledons of *E. sativa*.

The TEM observations of the DMSO + NaCl samples revealed significant precipitation of electron-dense bodies in the vacuole. Based on morphological similarity, we tentatively identified the electron-dense bodies as phenolic compounds (as described, for example, in [47,48]) or glucosinolates. An increase in phenolic content under salt stress during germination has been observed in lettuce and *Lonicera* [49,50], as well as in several legumes [51], and is interpreted as a protective response against salinity, possibly because enhanced phenolic synthesis may limit ROS generation by consuming reducing equivalents [31]. Salt stress has also been shown to induce glucosinolate accumulation during seed germination in *E. sativa* [52], and the biosynthetic pathways for phenols and at least some glucosinolates depend on the peroxidation of tyrosine and tryptophan [53,54]. The apparent absence of phenolic compounds in the presence of Rapamycin may therefore imply a reduction in salt-induced oxidative stress.

## 5. Conclusions

Rapamycin demonstrated the ability to restore the use of lipid droplets as a carbon source by cotyledon cells in germinating *Eruca sativa* seeds and seedlings exposed to salt stress. This effect may be interpreted as a protective response to salinity, associated with an increased rate of autophagic activity. The Rapamycin-induced autophagy was also linked to enhanced peroxisome turnover (as a result of pexophagy). The results presented here may be relevant for developing new strategies to improve germination rates and seedling survival in plants affected by salinity stress, thus representing an important connection between autophagy-related research and practical applications for crop growth improvement. Further investigations are needed to determine whether the increased autophagic activity, while enabling the reuse of lipid reserves in seeds, can also lead to improved growth under stress conditions. An analysis of cDNA transcripts related to stress and autophagy may provide crucial insights into the mechanism by which autophagy influences cellular metabolism under saline stress.

## Figures and Tables

**Figure 1 cells-14-01083-f001:**
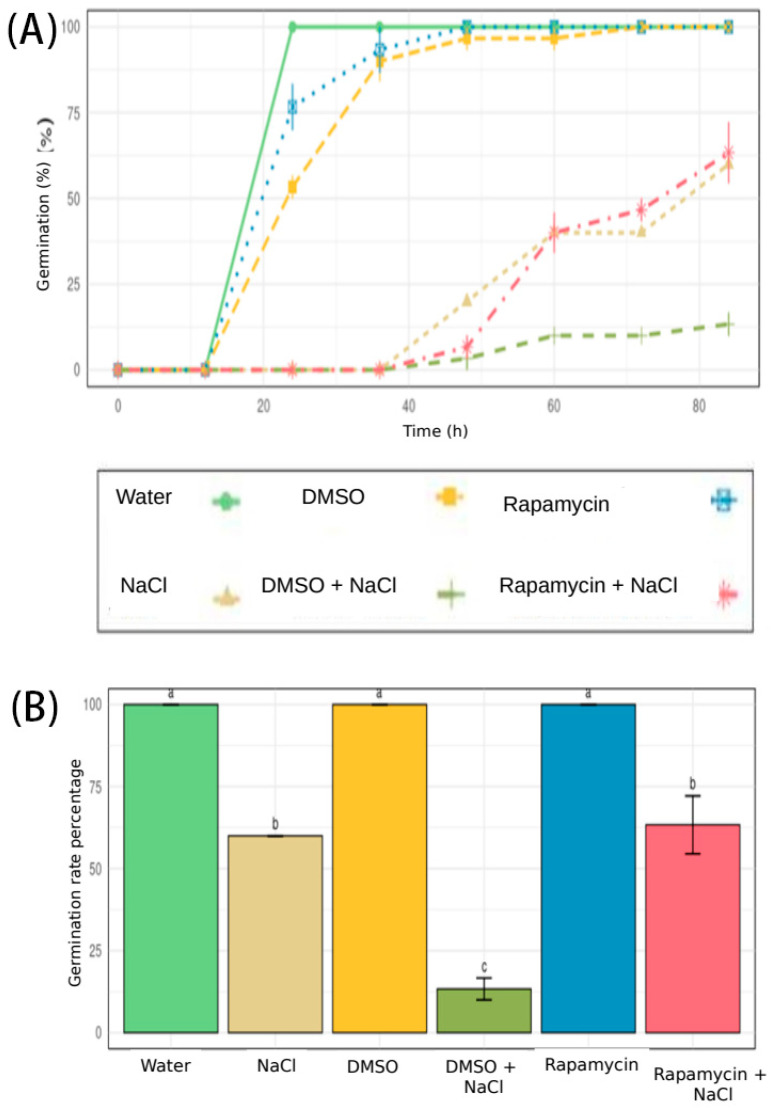
Percentage of germinated seeds over time. (**A**) Germination rates (lines) up to 72 h; (**B**) germination rate percentage (GRP) after 72 h shown as bars for *E. sativa* under various treatments. Distilled water, used as a general control (light green); NaCl 1.2% *w*/*v* (light brown), DMSO 0.2% *v*/*v*, used as experiment-relative control (yellow); DMSO 0.2% *v*/*v* and NaCl 1.2% *w*/*v* (dark green); Rapamycin 5 µM (blue); Rapamycin 5 µM + NaCl (pink). Different letters above the bars indicate significant differences (*p* < 0.05) among treatments.

**Figure 2 cells-14-01083-f002:**
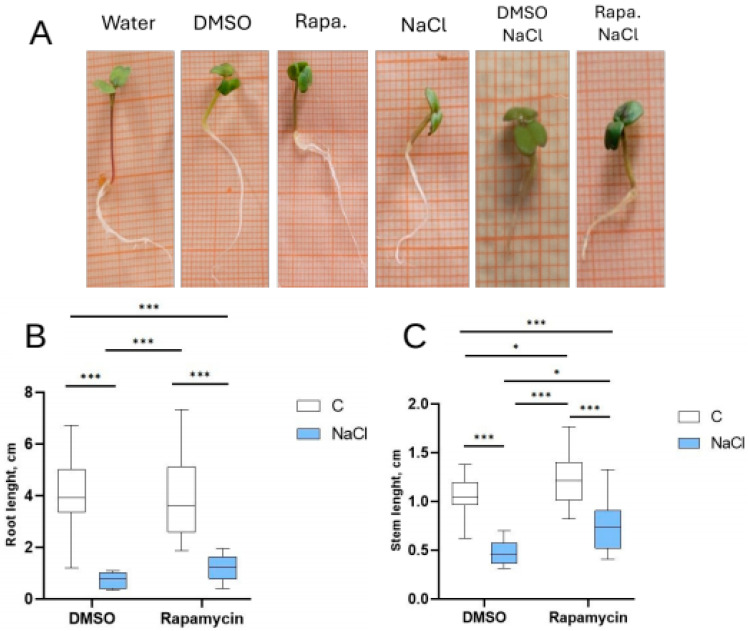
Effects on seedling growth. (**A**) Photographs of the seedlings grown with Rapamycin + NaCl compared to the controls (water alone and a solution of DMSO only) on graph paper as a dimensional reference (small squares with a side = 1 mm, and larger squares with a side = 5 mm). Effect of NaCl on the length of both the root (**B**) and stem (**C**) in *E. sativa* seedlings collected after 72 h. The number of asterisks is proportional to the level of significance (one asterisk means *p* < 0.05; three asterisks *p* ≤ 0.001).

**Figure 3 cells-14-01083-f003:**
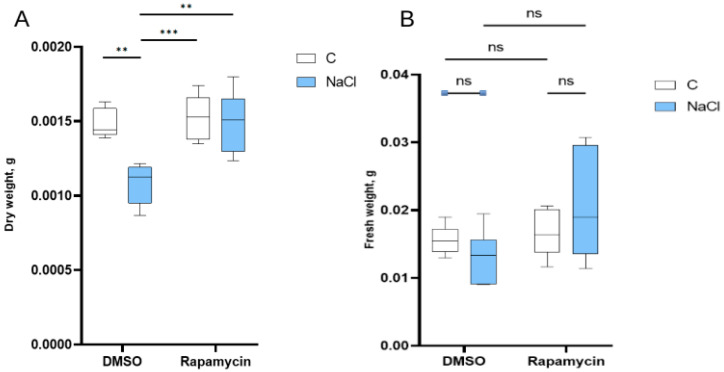
Effect of NaCl on the dry (**A**) and fresh (**B**) weights of *E. sativa* seedlings collected after 72 h. The number of asterisks is proportional to the level of significance (two asterisks means *p* ≤ 0.01; three asterisks *p* ≤ 0.001). C = control. ns = not significant.

**Figure 4 cells-14-01083-f004:**
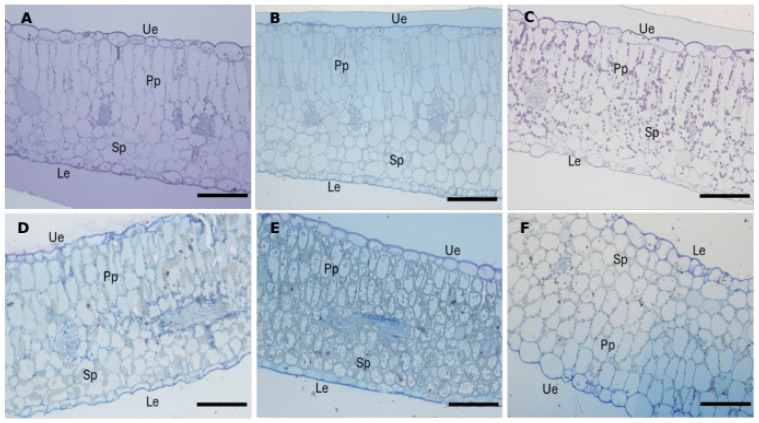
Semithin sections of *E. sativa* cotyledons stained with toluidine blue. (**A**) Cotyledon section of a water-treated seedling. (**B**) Cotyledon section of a DMSO-treated seedling. (**C**) Cotyledon section of a Rapamycin-treated seedling. (**D**) Cotyledon section of a NaCl-treated seedling. (**E**) Cotyledon section of a DMSO + NaCl-treated seedling. (**F**) Cotyledon section of a Rapamycin + NaCl-treated seedling. Ue = upper epidermis; Le = lower epidermis; Pp = palisade parenchyma; Sp = spongy parenchyma. Scale bar = 100 µm.

**Figure 5 cells-14-01083-f005:**
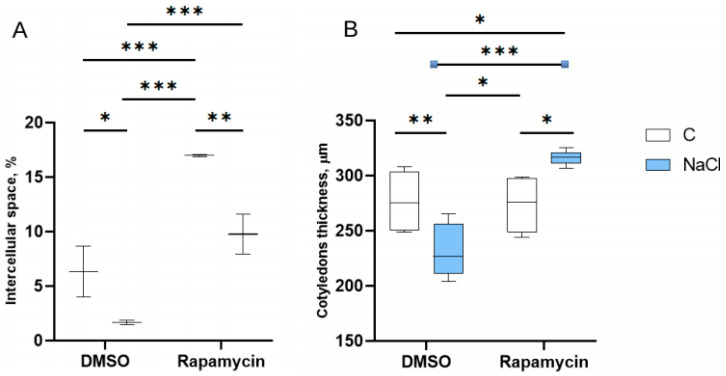
Effects on the size of the intercellular spaces. (**A**) Intercellular spaces (percentage of the area occupied in the section) of the mesophyll of plants treated with DMSO only vs. NaCl (with DMSO) and Rapamycin only vs. Rapamycin + NaCl. (**B**) Mesophyll thickness of cotyledons from plants under the different treatments. The number of asterisks is proportional to the level of significance (one asterisk means *p* < 0.05; two asterisks *p* ≤ 0.01; three asterisks *p* ≤ 0.001).

**Figure 6 cells-14-01083-f006:**
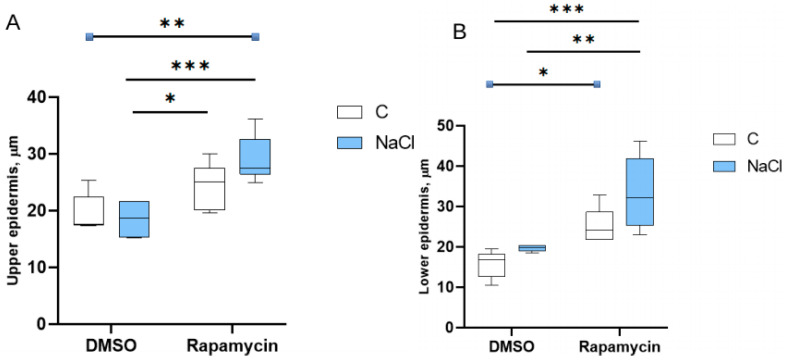
Effects on epidermis thickness. Thickness (µm) of the upper epidermis (**A**) and lower epidermis (**B**) of the cotyledons under the different treatments. The number of asterisks is proportional to the level of significance (one asterisk means *p* < 0.05; two asterisks *p* ≤ 0.01; three asterisks *p* ≤ 0.001).

**Figure 7 cells-14-01083-f007:**
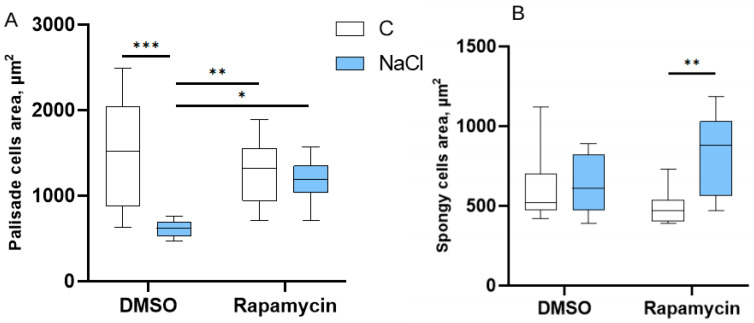
Effects on the mesophyll. Area of the palisade parenchyma cells (in µm^2^) (**A**) and spongy parenchyma cells (**B**) of the mesophyll of cotyledons under the different treatments. The number of asterisks is proportional to the level of significance (one asterisk means *p* < 0.05; two asterisks *p* ≤ 0.01; three asterisks *p* ≤ 0.001).

**Figure 8 cells-14-01083-f008:**
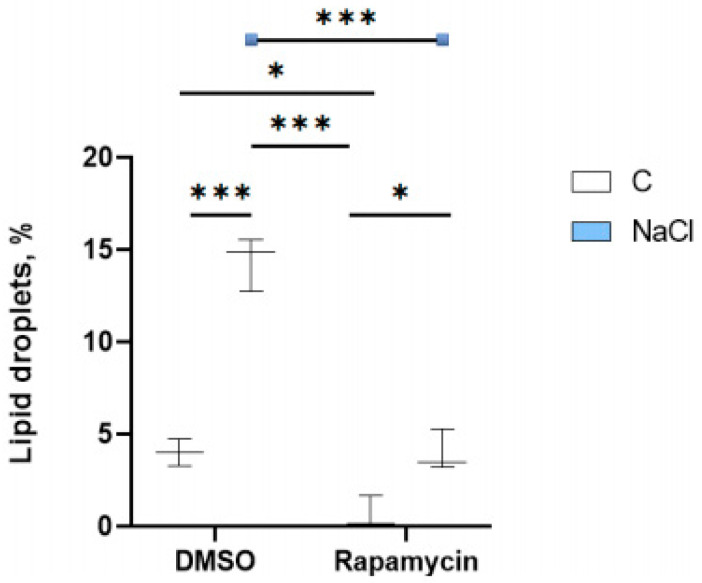
Percentage of the mesophyll area occupied by intracellular lipid droplets under the different treatments. The number of asterisks is proportional to the level of significance (one asterisk means *p* < 0.05; three asterisks *p* ≤ 0.001).

**Figure 9 cells-14-01083-f009:**
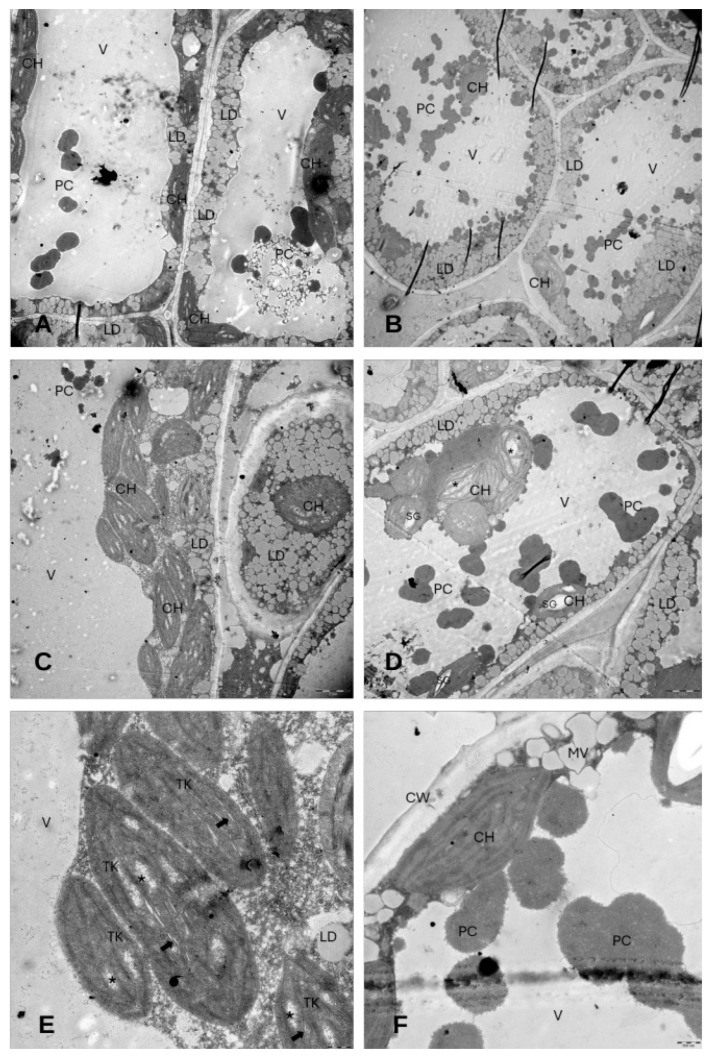
Electron micrographs of mesophyll cells in the leaves of *E. sativa* seedlings treated with DMSO + NaCl. (**A**,**B**) Overview of palisade cells and spongy cells showing the presence of several lipid droplets within the cytoplasm and electron-dense bodies similar to phenolic compounds inside the vacuole. (**C**,**D**) Magnification of mesophyll cells highlighting the high occurrence of lipid droplets and electron-dense bodies similar to phenolic compounds. (**E**) Detail of a chloroplast showing areas with mildly dilated thylakoids (black arrow), severely dilated thylakoids, and areas with less electron-dense stroma (asterisk). (**F**) Details of phenolic compounds within the vacuole. CH: chloroplast; LD: lipid droplet; V: vacuole; PC: electron-dense bodies similar to phenolic compounds; TK: thylakoid; SG: starch grain; MV: micro-vacuole. Figure (**A**–**D**) scale bar = 2 µm. Figure (**E**,**F**) scale bar = 500 nm.

**Figure 10 cells-14-01083-f010:**
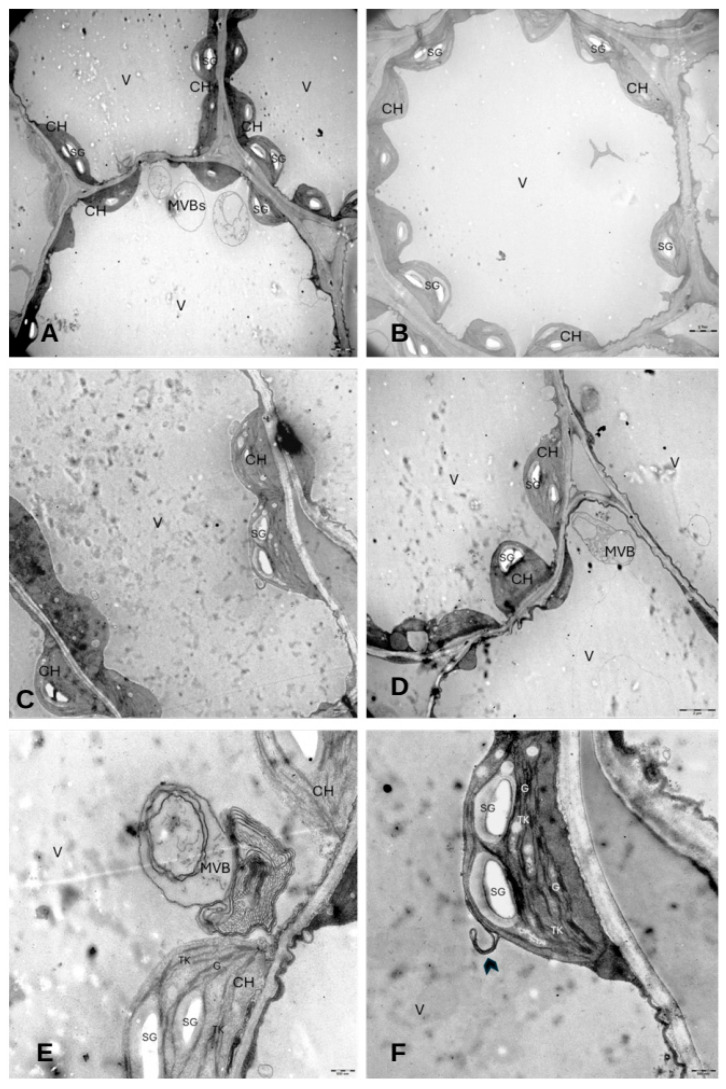
Electron micrographs of mesophyll cells in the leaves of *E. sativa* plants treated with Rapamycin + NaCl. (**A**,**B**) General view of palisade cells and spongy cells showing no lipid droplets within the cytoplasm and a clear vacuole. (**C**,**D**) Magnification of mesophyll cells highlighting chloroplasts with starch grains and empty vacuoles, if not for some MVBs. (**E**) Details of a chloroplast showing starch grains, a well-shaped membrane system, and a near MVB. (**F**) Details of a well-shaped chloroplast showing starch grains inside with a probable forming plastid-derived cisterna nearby (arrowhead). CH: chloroplast; LD: lipid droplet; V: vacuole; PC: electron-dense bodies similar to phenolic compounds; TK: thylakoid; SG: starch grain; MVB: multivesicular body. Figure (**A**–**D**) scale bar = 2 µm. Figure (**E**,**F**) scale bar = 500 nm.

**Figure 11 cells-14-01083-f011:**
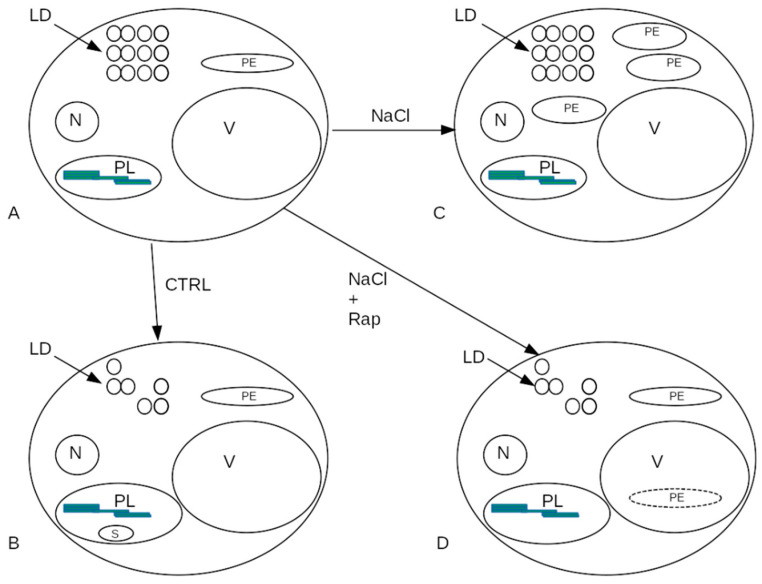
Effects of salt stress and Rapamycin on *E. sativa* seed germination. (**A**). Germination initiation. (**B**). Control: normal consumption of lipid reserves. (**C**). Effect of salt stress: failure to utilize lipid reserves. (**D**). Salt stress + Rapamycin: recovery of lipid droplet catabolism. N = nucleus; V = vacuole; PL = plastid; PE = peroxisome/glyoxysome; LD = lipid droplet; Rap = Rapamycin.

## Data Availability

The original contributions presented in this study are included in the article. Further inquiries can be directed to the corresponding author.

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
