# Peer review of "Rapamycin-Reactivated Lipid Catabolism in Eruca sativa Mill. Exposed to Salt Stress"

_cells, 2025, doi:10.3390/cells14141083_

Round 1

Reviewer 1 Report

Comments and Suggestions for Authors

This manuscript demonstrated that rapamycin alleviated the effects of salt stress on seed germination and seedling growth of arugula by inducing autophagy. However, the article lacks depth of mechanistic studies (lack of molecular/biochemical evidence to support them), the experimental design has control deficiencies, and some of the conclusions over-interpret the data. some problem:

  1. Based on the experimental findings, it is known that DMSO as a solvent can exacerbate salt stress damage, which then indicates that DMSO has a negative effect. However, there is no difference between DMSO and water in the experimental results presented by the authors. (For example, Figure 1…). In addition, the authors also mentioned in abstract “DMSO as a solvent does not appear to have a neutral effect on plant metabolism.” I think the authors should first clarify whether DMSO itself is specific, whether it has a direct effect on plant growth or metabolism, and how it exacerbates salt stress damage in plants during salt stress. In addition, the authors should explain in the introduction section what research has been done on DMSO in plants or other areas.
  2. Please add the basis for the selection of some of the experimental treatments covered in “2.1”, and the concentration combinations etc. applied by the experimental. Are the concentrations of Rapamycin and DMSO optimized by gradients? High concentrations of DMSO may be inherently toxic and the basis for selection needs to be explained.
  3. Please revise the title of 3.1. The title ‘Germination data’ is too general and not professional enough.
  4. When describing the results, it should be clear who is the control in the experimental treatments, rather than describing them one by one. For example, 3.1, 3.2. Please revise similar issues.
  5. “Figure” or “Fig”, please standardize the format of citations to figures.
  6. Each Figure legend should contain a concise title summarising the core content of the figure.
  7. Figure 3, What does the ‘C’ stand for in the illustration? Please make this clear in the legend.
  8. Figure 4, unavailable for viewing.
  9. Figure 8, the shape of the box-and-line diagram is basically invisible, and it is recommended that it be represented by other types of diagrams to make it more intuitive and aesthetic.
  10. In conclusion, “Rapamycin showed a protective effect on germinating seeds and seedlings in Eruca sativa exposed to salinity stress, be recovery of the capability to use lipid droplets via Glyoxylate cycle”. There is no direct evidence, and the authors should add biochemical or molecular experiments, such as enzyme activity assays, or expression of key genes.
  11. Please check the references section and make corrections.

Author Response

Response to Reviewer 1 Comments

1. Summary

We appreciated the effort by reviewer 1 to help us in improving our manuscript. We answered  each point and modified the manuscript accordingly. Please find  detailed responses below and the corresponding revisions/corrections highlighted in track changes mode in the re-submitted files. 

Reviewer’s 1 comments and answers
This manuscript demonstrated that rapamycin alleviated the effects of salt stress on seed germination and seedling growth of arugula by inducing autophagy. However, the article lacks depth of mechanistic studies (lack of molecular/biochemical evidence to support them), the experimental design has control deficiencies, and some of the conclusions over-interpret the data. some problem:
-Answ. We agree that the molecular/biochemical evidence is incomplete. We think that these other aspects could be treated in a separate article, since the morpho-physiological results here provided are already quite abundant. However, to improve the manuscript according to the reviewer 1 comment, we added literature about molecular evidence, specifically about transcription, about which there has been already contributions by other authors (about salt stress) and inserted an indication in the conclusion.

1) Based on the experimental findings, it is known that DMSO as a solvent can exacerbate salt stress damage, which then indicates that DMSO has a negative effect. However, there is no difference between DMSO and water in the experimental results presented by the authors. (For example, Figure 1…). In addition, the authors also mentioned in abstract “DMSO as a solvent does not appear to have a neutral effect on plant metabolism.” I think the authors should first clarify whether DMSO itself is specific, whether it has a direct effect on plant growth or metabolism, and how it exacerbates salt stress damage in plants during salt stress. In addition, the authors should explain in the introduction section what research has been done on DMSO in plants or other areas.
-answ. We agree with the reviewer that we treated these aspects in an insufficient way. DMSO is cited only because it is an excipient of Rapamycin and we probably dedicated too much space to that. We wanted only to explain why we used it in the control: it is because when we treat with Rapamycin we treat with DMSO too because it is provided together with this excipient. We explained better this point in the materials and methods. We omitted the sentence in the abstract and we clarified better in the text, also adding as requested some data about studies about the effects of DMSO on plants, but in materials and methods and discussion.
3) Please add the basis for the selection of some of the experimental treatments covered in “2.1”, and the concentration combinations etc. applied by the experimental. Are the concentrations of Rapamycin and DMSO optimized by gradients? High concentrations of DMSO may be inherently toxic and the basis for selection needs to be explained.
-answ. We explained better in the materials and methods: we used the DMSO concentration supplied together with Rapamycin.
4) Please revise the title of 3.1. The title ‘Germination data’ is too general and not professional enough.
-answ. We completely agreed and modified to “Effects on seed germination”
5) When describing the results, it should be clear who is the control in the experimental treatments, rather than describing them one by one. For example, 3.1, 3.2. Please revise similar issues.
-answ. Probably the misunderstanding derived from the presence of two controls, that is simply water and a DMSO solution. We explained better this point in the article. In several points the text was misleading, but I hope now it is clearer.
6) “Figure” or “Fig”, please standardize the format of citations to figures.
Answ. Done. I left “figure” in the first mention of this term
7) Each Figure legend should contain a concise title summarising the core content of the figure.
-Answ. Done
8) Figure 3, What does the ‘C’ stand for in the illustration? Please make this clear in the legend.
-Answ. Done. It was control. We apologize for having omitted the explanation.
9) Figure 4, unavailable for viewing.
-Answ. Probably some problems with the conversion to PDF. Fig. 4 is now in place.
10) Figure 8, the shape of the box-and-line diagram is basically invisible, and it is recommended that it be represented by other types of diagrams to make it more intuitive and aesthetic.
-Answ. Done. We improved the image by encircling it inside a box
11) In conclusion, “Rapamycin showed a protective effect on germinating seeds and seedlings in Eruca sativa exposed to salinity stress, be recovery of the capability to use lipid droplets via Glyoxylate cycle”. There is no direct evidence, and the authors should add biochemical or molecular experiments, such as enzyme activity assays, or expression of key genes.
-Answ. We changed the conclusion stating that Rapamycin is able to reestablish the use of lipid droplets
12) Please check the references section and make corrections.
-Answ. done

Reviewer 2 Report

Comments and Suggestions for Authors

The manuscript focuses on investigating the effect of rapamycin, an mTOR inhibitor and autophagy inducer, on salt toxicity in Eruca sativa. The authors observed that salt stress in the presence of DMSO inhibited the catabolism of lipid droplets stored in the embryo of E. sativa. Consequently, lipid droplets persisted in the developing seedling until a late stage. In contrast, lipid catabolism under salt stress combined with rapamycin treatment was comparable to the control condition. Rapamycin-induced autophagic activity was demonstrated using Transmission Electron Microscopy, showing increased membrane recycling.

This study addresses an interesting and scientifically relevant topic. Therefore, I recommend its publication after minor revisions.

To enhance the scientific rigor and improve the formal structure of the manuscript, I suggest the following revisions:

  • The authors should provide a clear and detailed justification for the selected concentrations of rapamycin, NaCl, and DMSO. It is important to clarify whether these concentrations are based on previous literature, physiological relevance, or preliminary experiments. Without this information, assessing the validity and broader applicability of the findings is challenging.
  • The research objective should be more clearly and comprehensively articulated to improve the manuscript’s focus and clarity.
  • The expressions “salt-induced sufference” and “suffered by the plant” are inappropriate in scientific writing and should be replaced with more formal, and precise terminology.
  • The conclusions section should be expanded and strengthened. Currently, it reads as a summary or repetition of the results rather than offering a critical interpretation. The authors are encouraged to emphasize the broader significance of their findings and suggest potential directions for future research.
  • A minor revision of the English language is recommended to improve clarity and overall readability throughout the manuscript.

Author Response

-Answ. We thank the reviewer for the very useful suggestions
The manuscript focuses on investigating the effect of rapamycin, an mTOR inhibitor and autophagy inducer, on salt toxicity in Eruca sativa. The authors observed that salt stress in the presence of DMSO inhibited the catabolism of lipid droplets stored in the embryo of E. sativa. Consequently, lipid droplets persisted in the developing seedling until a late stage. In contrast, lipid catabolism under salt stress combined with rapamycin treatment was comparable to the control condition. Rapamycin-induced autophagic activity was demonstrated using Transmission Electron Microscopy, showing increased membrane recycling.
This study addresses an interesting and scientifically relevant topic. Therefore, I recommend its publication after minor revisions.
To enhance the scientific rigor and improve the formal structure of the manuscript, I suggest the following revisions:
    • The authors should provide a clear and detailed justification for the selected concentrations of rapamycin, NaCl, and DMSO. It is important to clarify whether these concentrations are based on previous literature, physiological relevance, or preliminary experiments. Without this information, assessing the validity and broader applicability of the findings is challenging. 
-Answ. We better explained these points in the “Materials and methods” part. We used the DMSO  concentration at which this substance is provided together with Rapamycin, Rapamycin concentration was chosen after literature and NaCl from previous experiments by the authors. Everything is now explained in the methods.
    • The research objective should be more clearly and comprehensively articulated to improve the manuscript’s focus and clarity. 
    • -Answ. We added a better explanation of the objective at the end of the introduction
    • The expressions “salt-induced sufference” and “suffered by the plant” are inappropriate in scientific writing and should be replaced with more formal, and precise terminology. 
    • -Answ. Done. We replaces sufference with stress and  growth impairment
    • The conclusions section should be expanded and strengthened. Currently, it reads as a summary or repetition of the results rather than offering a critical interpretation. The authors are encouraged to emphasize the broader significance of their findings and suggest potential directions for future research. 
    • -Answ. We agree and  we improved this part accordingly
    • A minor revision of the English language is recommended to improve clarity and overall readability throughout the manuscript. 
    • -Answ. We improved the english language

Reviewer 3 Report

Comments and Suggestions for Authors

The current paper the effect of Rapamycin was investigated on Eruca sativa L. seedlings under salt stress conditions. Authors investigated leaf anatomy and cell ultrastructure under control and under salt stress conditions.

The authors made a good work and use many modern methods.

However, revision and clarifications are required.

Details:

Line 9: „autophagy inducer on salt toxicity in Eruca sativa.“ ??

Line 13: “We observed an increase in salt-induced suffering in germinating seeds and seedlings in E. sativa in the presence of DMSO, the solvent normally provided with Rapamycin.“ ?? This message is unclear, Mix two different points.

Line 23: „would reactivate,the Glyoxylate cycle..“ ?? double points, unnecessary comma…

Line 31: „reduced availability of water“ ??? The authors need to describe the actual mechanism of salt (sodium) toxicity involves vacuolar alkalinization or the prevention of vacuolar acidification. During the growth cycle, vacuoles normally undergo acidification, which facilitates water uptake and supports cell expansion. However, the accumulation of sodium in the vacuole interferes with this process, preventing proper cell expansion. This disruption leads to an imbalance between different cell types, ultimately resulting in reduced plant productivity.

Line 40: „salt stress damages plant cell structure“ ??? Stress can not damaged structure itself.

Line 50: layout/double space.

Line 76: „with 1 mL of treatment solution“?? I am not sure that 1 ml is enough oper such a large surface. How did you keep moisture? How rapid this 1 ml were evaporated?

Have you use wett chamber?

Line 82: „72 h (5 days)“ ?? 5d = 120 h!  I am not sure root growth is a real marker of germination. The real marker is chormatin remodelling.

Line 101: „Inserire qui le misure degli spessori etc con Image J.” ¿??

Figure 1: please, insert panel label A, B. Explain what do you mean as “germination %“, at which time point?

Line 148: clarify day, please „at the end of the experiment“.

Line 150: „subjected, Rapamycin + NaCl“ ?? Layout!

Figure 2: images is too small, no scale bar.

Lines 214- 216- why empty space?

Line 234: „showed a value of the area occupied” ¿?

Gigure 8: please, improve layout. Maybe column will look better as lines.

Line 299: „the leaves“ ?? Maybe cotyledon? Or true leaf?

Discussion: citation layout! Figure 11: layout.

The DMSO have some effect in concetntarion high as 1 µl/ml. Here you used 2 µl/ml. So, the results is clar. Please, next time use 2-4 times more concentrated stock.

Moreover, hypocotyl/stem length and root length during germination is dependent from carbon supply. In precitice here you study „carbon starvation“ and this need to be mention in discussion.

Please, check carefully whole text, adjust layout , grammar, figures.   

Comments on the Quality of English Language

major correction are required. See comments.

Author Response

-Answ. We thank the reviewer for the very useful suggestions
The current paper the effect of Rapamycin was investigated on Eruca sativa L. seedlings under salt stress conditions. Authors investigated leaf anatomy and cell ultrastructure under control and under salt stress conditions.
The authors made a good work and use many modern methods.
However, revision and clarifications are required.
Details:
Line 9: „autophagy inducer on salt toxicity in Eruca sativa.“ ??
-Answ. corrected

Line 13: “We observed an increase in salt-induced suffering in germinating seeds and seedlings in E. sativa in the presence of DMSO, the solvent normally provided with Rapamycin.“ ?? This message is unclear, Mix two different points.
-Answ. We agree and eliminated the sentence 
Line 23: „would reactivate,the Glyoxylate cycle..“ ?? double points, unnecessary comma…
-Answ. corrected
Line 31: „reduced availability of water“ ??? The authors need to describe the actual mechanism of salt (sodium) toxicity involves vacuolar alkalinization or the prevention of vacuolar acidification. During the growth cycle, vacuoles normally undergo acidification, which facilitates water uptake and supports cell expansion. However, the accumulation of sodium in the vacuole interferes with this process, preventing proper cell expansion. This disruption leads to an imbalance between different cell types, ultimately resulting in reduced plant productivity.
-Answ. We overlooked this point, and we are grateful  for giving us the opportunity to edit it. We added a new reference in relation to this point
Line 40: „salt stress damages plant cell structure“ ??? Stress can not damaged structure itself.
-Answ. We agree. We changed to “influences”
Line 50: layout/double space
-Answ. .corrected 
Line 76: „with 1 mL of treatment solution“?? I am not sure that 1 ml is enough oper such a large surface. How did you keep moisture? How rapid this 1 ml were evaporated?
Have you use wett chamber?
-Answ. We needed to clarify this point: we used chambers where humidity is  satured and it was 1 mL per seed.  We clarified the point in materials and methods

Line 82: „72 h (5 days)“ ?? 5d = 120 h!  I am not sure root growth is a real marker of germination. The real marker is chormatin remodelling.
-Answ. You are right.  We needed to prolong the experiment because some of the seeds treated with high salt concentration had a delayed response. 
Line 101: „Inserire qui le misure degli spessori etc con Image J.” ¿??
-Answ. Sorry: a typos derived by unfortunate recovery of a previously eliminated sentence. Fixed. Thank You
Figure 1: please, insert panel label A, B. Explain what do you mean as “germination %“, at which time point?
-Answ. done
Line 148: clarify day, please „at the end of the experiment“.
-Answ. done
Line 150: „subjected, Rapamycin + NaCl“ ?? Layout!
-Answ. Done We apologize for the error.
Figure 2: images is too small, no scale bar.
-Answ. Done. Image increased Probably the dimension was reduced during the conversion to pdf. Explanation in the legend about the use of graph paper, hence the lack of the bar
Lines 214- 216- why empty space? 
-Answ. Done. That came out by trying to position the images correctly. We corrected
Line 234: „showed a value of the area occupied” ¿?
-Answ. Done. It was the area occupied in the section
Gigure 8: please, improve layout. Maybe column will look better as lines.
-Answ. Done. We tried to improve the image
Line 299: „the leaves“ ?? Maybe cotyledon? Or true leaf?
-Answ. corrected. 
Discussion: citation layout! Figure 11: layout.
-Answ. corrected. 
The DMSO have some effect in concetntarion high as 1 µl/ml. Here you used 2 µl/ml. So, the results is clar. Please, next time use 2-4 times more concentrated stock.
-Answ. We agree: it is not easy since Rapamycin stock is provided with DMSO at a given concentration and diluting DMSO implies diluting Rapamycin too. 
Moreover, hypocotyl/stem length and root length during germination is dependent from carbon supply. In precitice here you study „carbon starvation“ and this need to be mention in discussion.
-Answ. I inserted a reference to this concept after treating the insufficient capability of using lipids as carbon source. 
Please, check carefully whole text, adjust layout , grammar, figures.   
Answ. We rechecked everything hoping not to have overlooked anything

Reviewer 4 Report

Comments and Suggestions for Authors

This study investigates the effect of the mTOR inhibitor Rapamycin on seed germination, seedling growth, and cellular metabolism in Eruca sativa (rocket) under salt stress. The experimental design is well-structured, the data presented is reasonably complete, and the conclusions demonstrate novelty, contributing significantly to our understanding of plant salt stress responses and autophagy regulatory mechanisms. However, certain aspects require further clarification, methodological details need supplementation, and the text requires optimization. Specific comments are outlined below:

  1. Introduction:â‘ Supplement Research Background: The background on Rapamycin's role in plants needs expansion. Specifically, highlight existing research on Rapamycin and the TOR pathway in plants to clearly position the innovative aspects of this study (e.g., focus on lipid catabolism reactivation via pexophagy under salt stress).

  1. Materials and Methods:â‘  Section 2.3 (ImageJ Analysis): The placeholder text "Inserire qui le misure degli spessori etc con Image J." indicates missing details. Must supplement with a clear description of how ImageJ was used to quantify parameters like thicknesses, intercellular space area, lipid droplet percentage area, palisade/spongy mesophyll areas, etc. (e.g., thresholding methods, measurement tools used, sampling strategy per section).
  2. Results:â‘  Section 3.1 (Germination data): While the statistical graph (Fig. 1) is provided, readers would benefit significantly from seeing representative phenotypic images of germinating seeds/seedlings for each treatment group at key time points, alongside the graph.â‘¡ Figures 2 and 4 are currently missing/unavailable in the provided text. These must be included for a proper evaluation of the results (Fig. 2: Seedling morphology; Fig. 4: Light microscopy of cotyledons).â‘¢ Statistical Significance (Figs 5, 6, 7, 8): The text describing the results for these figures does not explicitly state which specific treatment groups are being compared when reporting significant differences (denoted by asterisks in the figures). This information must be clearly provided in the results text corresponding to each figure (e.g., "Treatment X was significantly different from Treatment Y (p<0.05)"). â‘£ Quantification Needs: Germination Delay (Section 3.1): Quantify the observed delay. E.g., "Germination initiation in the NaCl group was delayed by approximately XX hours compared to the control." â‘¤ Lipid Droplet Percentage (Section 3.4 / Fig 8): Merely stating "significant increase" is insufficient. Provide specific numerical values or fold-changes (e.g., "The area occupied by lipid droplets increased from X% in the control to Y% in the DMSO+NaCl group"). â‘¥ TEM Image Annotation (Section 3.9 / Figs 9 & 10): Key structures described in the text (e.g., Lipid Droplets (LD), Multi Vesicular Bodies (MVBs), phenolic compounds (PC), chloroplasts (CH), vacuoles (V), starch grains (SG)) must be clearly indicated on the figures themselves using arrows and labels.

  1. Discussion:â‘  DMSO Negative Effect: The discussion on the detrimental effect of DMSO under salt stress needs deepening regarding the potential mechanism(s). Explore possibilities beyond simple solvent presence, such as whether DMSO exacerbates osmotic stress or causes membrane damage/integrity disruption, contributing to the observed lipid catabolism blockage. â‘¡ Phenolic Compounds & Lipid Metabolism: The observation of phenolic compound accumulation under salt stress (DMSO+NaCl, Fig 9) is noted, but the discussion fails to explore any potential link between this phenomenon and the impairment of lipid metabolism. Discuss possible connections, such as increased antioxidant demand under stress potentially diverting resources or signaling pathways.

  1. Formatting & Language:â‘  Abbreviation: "Transmission Electron Microscope" is used repetitively in the Abstract. Replace subsequent uses after the first full mention with the standard abbreviation TEM.â‘¡ Line 22: "reactivate,the Glyoxylate cycle" contains an erroneous comma. Correct to: "reactivate the Glyoxylate cycle". â‘¢ English Editing: The overall manuscript requires thorough English language editing by a native speaker or professional service to ensure grammatical accuracy, clarity, fluency, and consistent scientific terminology (e.g., consistent use of peroxisomes/glyoxysomes).
Comments on the Quality of English Language

The overall manuscript requires thorough English language editing by a native speaker or professional service to ensure grammatical accuracy, clarity, fluency, and consistent scientific terminology (e.g., consistent use of peroxisomes/glyoxysomes).

Author Response

Reviewer 4’s comments and answers
This study investigates the effect of the mTOR inhibitor Rapamycin on seed germination, seedling growth, and cellular metabolism in Eruca sativa (rocket) under salt stress. The experimental design is well-structured, the data presented is reasonably complete, and the conclusions demonstrate novelty, contributing significantly to our understanding of plant salt stress responses and autophagy regulatory mechanisms. However, certain aspects require further clarification, methodological details need supplementation, and the text requires optimization. Specific comments are outlined below:

    1. Introduction:â‘ Supplement Research Background: The background on Rapamycin's role in plants needs expansion. Specifically, highlight existing research on Rapamycin and the TOR pathway in plants to clearly position the innovative aspects of this study (e.g., focus on lipid catabolism reactivation via pexophagy under salt stress). 
 -Answ. Thank You for this fundamental comment: we added information in the part dedicated to autophagic activity. We added two more citations
    2. Materials and Methods:â‘  Section 2.3 (ImageJ Analysis): The placeholder text "Inserire qui le misure degli spessori etc con Image J." indicates missing details. Must supplement with a clear description of how ImageJ was used to quantify parameters like thicknesses, intercellular space area, lipid droplet percentage area, palisade/spongy mesophyll areas, etc. (e.g., thresholding methods, measurement tools used, sampling strategy per section). 
 -Answ. Done. I rechecked also the site of ImageL
    3. Results:â‘  Section 3.1 (Germination data): While the statistical graph (Fig. 1) is provided, readers would benefit significantly from seeing representative phenotypic images of germinating seeds/seedlings for each treatment group at key time points, alongside the graph.
Answ: we can provide the images as supplementary material
â‘¡ Figures 2 and 4 are currently missing/unavailable in the provided text. These must be included for a proper evaluation of the results (Fig. 2: Seedling morphology; Fig. 4: Light microscopy of cotyledons).
Answ. : I suppose it is a problem related to the conversion of the text file in pdf. I hope in the new version the images will be visible
â‘¢ Statistical Significance (Figs 5, 6, 7, 8): The text describing the results for these figures does not explicitly state which specific treatment groups are being compared when reporting significant differences (denoted by asterisks in the figures). This information must be clearly provided in the results text corresponding to each figure (e.g., "Treatment X was significantly different from Treatment Y (p<0.05)").
Answ. : done
â‘£ Quantification Needs: Germination Delay (Section 3.1): Quantify the observed delay. E.g., "Germination initiation in the NaCl group was delayed by approximately XX hours compared to the control." 
Answ. : done
⑤ Lipid Droplet Percentage (Section 3.4 / Fig 8): Merely stating "significant increase" is insufficient. Provide specific numerical values or fold-changes (e.g., "The area occupied by lipid droplets increased from X% in the control to Y% in the DMSO+NaCl group"). 
Answ. : done
â‘¥ TEM Image Annotation (Section 3.9 / Figs 9 & 10): Key structures described in the text (e.g., Lipid Droplets (LD), Multi Vesicular Bodies (MVBs), phenolic compounds (PC), chloroplasts (CH), vacuoles (V), starch grains (SG)) must be clearly indicated on the figures themselves using arrows and labels. 
Answ. : we improved lettering explanation in the figure legends and we did not insert to many arrows to avoid confusion with the lettering

    3. Discussion:â‘  DMSO Negative Effect: The discussion on the detrimental effect of DMSO under salt stress needs deepening regarding the potential mechanism(s). Explore possibilities beyond simple solvent presence, such as whether DMSO exacerbates osmotic stress or causes membrane damage/integrity disruption, contributing to the observed lipid catabolism blockage.
Answ. : we added some literature about the possible mechanism (increase of membrane permeability). We reduced a bit the part on DMSO
 â‘¡ Phenolic Compounds & Lipid Metabolism: The observation of phenolic compound accumulation under salt stress (DMSO+NaCl, Fig 9) is noted, but the discussion fails to explore any potential link between this phenomenon and the impairment of lipid metabolism. Discuss possible connections, such as increased antioxidant demand under stress potentially diverting resources or signaling pathways. 
Answ. : done. We improved this part with several literature entries

    4. Formatting & Language:â‘  Abbreviation: "Transmission Electron Microscope" is used repetitively in the Abstract. Replace subsequent uses after the first full mention with the standard abbreviation TEM.
Answ. : done
â‘¡ Line 22: "reactivate,the Glyoxylate cycle" contains an erroneous comma. Correct to: "reactivate the Glyoxylate cycle". 
Answ. : done (sentence changed)
â‘¢ English Editing: The overall manuscript requires thorough English language editing by a native speaker or professional service to ensure grammatical accuracy, clarity, fluency, and consistent scientific terminology (e.g., consistent use of peroxisomes/glyoxysomes). 
Answ. : we tried to improve this part also citing the definition of per/glyo

Comments on the Quality of English Language
The overall manuscript requires thorough English language editing by a native speaker or professional service to ensure grammatical accuracy, clarity, fluency, and consistent scientific terminology (e.g., consistent use of peroxisomes/glyoxysomes).
Answ. : Answ. : we revised with the help of a mother tongue

Round 2

Reviewer 1 Report

Comments and Suggestions for Authors

Unfortunately, the authors have refused to provide further supplementary data to provide more comprehensive evidence. The authors believe that “these other aspects could be treated in a separate article, since the morpho-physiological results here provided are already quite abundant.” 

The authors have appropriately revised the relevant conclusions to align with the existing experimental evidence. Furthermore, they have addressed the additional requested revisions. Based on these improvements, I believe the manuscript now meets the necessary standards and is acceptable for publication. Thank you for your consideration.

Author Response

We thank the reviewer for the comments. We will surely follow his suggestions to proceed on this subject to provide more evidence on the molecular side

Reviewer 3 Report

Comments and Suggestions for Authors

Thank you! The paper need some more corrections. 

Lines 55- 57 - layout (citation layout, space). Moreover, "sensitive to stress damage" ?? Sensitive to stress. Sensitive to damage is not correct.

"Pexophagy, the selective macroautophagy of peroxisomes, is the mechanism"  pexophagy formation can be the mechanism , not pexophygy itself.

Moreover, authors still not described mechnanism of salt stress: sodium ion prevent vacuaole growth in cell undergo specifci function, lead to imbalance between organneles and it is a reason of "damage", but not damage organelles itself. Peroxisomes increasing size during cell specifcication/differentiation (mesophyll cell as example), but with soidium cell can not growth in size, while peroxisomes can. It is the reason your observed,

Perhaps, you  overlooked a "butterfly effect" of the sodium and try to explain events you see by direct correlation.. Please, think about this in discussion and for future planing,      

Line 84: maybe you can oredr rapamycin as powder and prepare more concentrated stock to avoid excess DMSO in the medium? For the future.

Line 110: "2 mm long geotropic rootlet" ?? seeds germination is rather radicle come out from endosüperm/aleurone layers. 2mm  -is normal root growth because of cell explansion. Once root pass through endosperm, they start to irreversible use internal resources and can be consider as germinated. Please, take into account in discussion and in fiture plans. 

Line227: layout

Lines 252-254 - empty.

Line 295: layout.

Line 311: "phenol compounds" - how do you detect it?

Line 387 . 389: Huang et al: in which plants? cell types?  

Lines 402 - 427: layout.

Model: should be linked with cell type and shown effect of the salt stress on vacuole and cell size, with mechanosense "stress" on the other organelles. 

Line 111: "to 72 h (5 days). " - still wrong..

Line 164: "24 hours before beginning to start germination"??

Author Response

-Answ. : we thank the reviewer for the further suggestions

Lines 55- 57 - layout (citation layout, space). Moreover, "sensitive to stress damage" ?? Sensitive to stress. Sensitive to damage is not correct.

-Answ. : done

"Pexophagy, the selective macroautophagy of peroxisomes, is the mechanism"  pexophagy formation can be the mechanism , not pexophygy itself.

-Answ. : apparently other authors cite the term “pexophagy” as a mechanism. Probably it is still a matter of definition. Following the reviewer’s indication we corrected to process

Moreover, authors still not described mechnanism of salt stress: sodium ion prevent vacuaole growth in cell undergo specifci function, lead to imbalance between organneles and it is a reason of "damage", but not damage organelles itself. Peroxisomes increasing size during cell specifcication/differentiation (mesophyll cell as example), but with soidium cell can not growth in size, while peroxisomes can. It is the reason your observed,

-Answ. : this observation is an interesting starting point for further ideas. We studied more literature about the subject and modified the text accordingly

Perhaps, you  overlooked a "butterfly effect" of the sodium and try to explain events you see by direct correlation.. Please, think about this in discussion and for future planing,      

-Answ. : again: also this observation is an interesting starting point for further ideas. We treated it in the discussion [ingresso di acqua? Uscita di acqua? Da cell o orgenelle]

Line 84: maybe you can oredr rapamycin as powder and prepare more concentrated stock to avoid excess DMSO in the medium? For the future.

-Answ. : we agree, even if we used the same product previously employed in other investigations

Line 110: "2 mm long geotropic rootlet" ?? seeds germination is rather radicle come out from endosüperm/aleurone layers. 2mm  -is normal root growth because of cell explansion. Once root pass through endosperm, they start to irreversible use internal resources and can be consider as germinated. Please, take into account in discussion and in fiture plans. 

-Answ. : we agree: the choice of word was misleading and we rephrased it

Line227: layout

-Answ. : corrected

Lines 252-254 – empty.

-Answ. : corrected. They were inserted for positioning the figures in the text. In the last version i will provide the figures separately

Line 295: layout.

-Answ. :corrected. Same as above.

Line 311: "phenol compounds" - how do you detect it?

-Answ. :By comparison of morphological features in literature. I added another literature entry and we changed to similar to phenol compounds

Line 387 . 389: Huang et al: in which plants? cell types?  

-Answ. :It was in seedlings of Eruca after germination: we inserted this relevant information as suggested

.

Lines 402 - 427: layout.

Model: should be linked with cell type and shown effect of the salt stress on vacuole and cell size, with mechanosense "stress" on the other organelles. 

-Answ. :we tried to change accordingly

Line 111: "to 72 h (5 days). " - still wrong..

-Answ. : i am really sorry for the mistake. Corrected.

Line 164: "24 hours before beginning to start germination"??

-Answ. : we corrected. Thank You

Reviewer 4 Report

Comments and Suggestions for Authors

Accept. I think the authors have addressed the reviewers' comments very well.

Author Response

Thank You for Your help